# Rural populations exposed to Ebola Virus Disease respond positively to localised case handling: Evidence from Sierra Leone

Esther Yei Mokuwa[1], Harro Maat[2]*

1 Development Economics group, Wageningen University, Wageningen, The Netherlands, 2 Knowledge, Technology and Innovation group, Wageningen University, Wageningen, The Netherlands

* Harro.Maat@wur.nl

## Abstract

At the height of the Ebola epidemic in Sierra Leone in November 2014, a new decentralized approach to ending infection chains was adopted. This approach was based on building local, small-scale Community Care Centres (CCC) intended to serve as triage units for safe handling of patients waiting for test results, with subsequent transfer to Ebola Treatment Centers (ETC) for those who tested positive for Ebola. This paper deals with local response to the CCC, and explains, through qualitative analysis of focus group data sets, why this development was seen in a positive light. The responses of 562 focus group participants in seven villages with CCC and seven neighbouring referral villages without CCC are assessed. These data confirm that CCC are compatible with community values concerning access to, and family care for, the sick. Mixed reactions are reported in the case of "safe burial", a process that directly challenged ritual activity seen as vital to maintaining good relations between socially-enclaved rural families. Land acquisitions to build CCC prompted divided responses. This reflects problems about land ownership unresolved since colonial times between communities and government. The study provides insights into how gaps in understanding between international Ebola responders and local communities can be bridged.

**Data Availability Statement:** Data are part of a larger data set as part of a PhD project. Data used for this article are made available as attachments to the paper (see S1 File).

## Author summary

Control of Ebola Virus Disease requires facilities where patients can be isolated and treated safely, without risk to medical personnel or family members. In the 2014–15 Ebola epidemic in Sierra Leone emphasis was at first placed on large field hospitals known as Ebola Treatment Centers (ETC). These were often located far from areas where new cases were being discovered. Patients were distrustful of their purpose and slow to report, and the disease continued to spread. Six months into the epidemic a new approach was tried, based on much smaller and more rapidly constructed centres (Community Care Centres (CCC) located where new cases were occurring. This paper examines community reactions to the CCC. There was a much greater sense of community ownership of these small, localised centres, and reporting times improved. Families were able easily to visit

**Funding:** The author(s) received no specific funding for this work

**Competing interests:** No authors have competing interests.

and observe activities, even though restricted from crossing red lines. The staff were often local and provided trustworthy information on the progress of patients. Families were able to prepare home food for patients, and this was thought to improve their morale and chances of survival. CCC were also appreciated for treating other disease, and not only Ebola. Referral of patients to ETC was easier to accept when the outcome of an Ebola blood test was known. There were some differences of opinion over "safe burial" procedures and acquisition of sites for the CCC, but on balance CCC were well accepted by communities, and were seen locally as a positive development in Ebola control.

## Introduction

In the epidemic of 2014–15 Sierra Leone had a total of 8630 laboratory confirmed cases of Ebola Virus Disease (EVD) [1]. The international community constructed Ebola Treatment Centers (ETC) as a key part of the epidemic response. These were facilities with very strict bio-safety control, capable of handling 100 or more cases at a time. The International Federation of Red Cross and Red Crescent Societies (IFRC) opened such a facility at Nganyahun, about ten miles north of Kenema, in September 2014, followed by other units in Bo, Freetown and Makeni (all urban locations).

   ETC were initially viewed by communities as distant, hostile places where patients went to die. Families feared patients would be forcibly carried to such a facility. Lack of sufficient bed capacity as the epidemic peaked, coupled with community unease, led to a modified approach–the building of a series of 55 small-scale Community Care Centres (CCC), beginning in November 2014. They were established in high-transmission areas, far from the capital and ETCs, with poor transport access. Existing rural health facilities (public health units, PHU) were felt to be insufficiently 'protective' in terms of infection control to house Ebola patients. Additionally, many villages were afraid to use health centers due to fear of contracting Ebola. Some CCCs were located on PHU grounds–and were intended to allow 'routine' cases (non-Ebola suspects) more safely to use PHU facilities. Average build time was about two weeks, and CCC were staffed with local medical and non-medical staff and with some international volunteers.

   Although the best strategy might be to isolate and test all suspect cases in ETC as quickly as possible (within the first three days of onset of high fever symptoms) this was undermined, in Sierra Leone, by patient resistance and lack of capacity. Fear of ETC led to hiding of patients [1]. Shortage of beds also hampered Ebola response. By October 2014 there were only 287 beds in four ETC, all located in urban centres [2]. Locations (notably Bo and Kenema) were far from the places where new cases were occurring (in Freetown, Kono and the north). Care *in situ* was considered but rejected. Giving families Personal Protective Equipment (PPE) to minimize transmission while nursing patients at home would be problematic. Safe use of PPE was difficult even for professionals, and the country lacked enough trainers to instruct families in relevant nursing skills [3].

   At the request of the government and with endorsement from WHO, leaders of the response to EVD in Sierra Leone decided to support another approach–passive case finding with community isolation. Those with suspected EVD would be encouraged to gather in units where they would receive basic care, and avoid infecting their families [3]. It was reasoned that many small units would be better than a few large ones, since they could be placed closer to emergent hotspots of a disease that moved in complex, non-linear jumps [4]. The original plan was to build up to 200 CCCs though in the end only 55 units were needed, in 5 districts in

Sierra Leone, due to downturn in numbers of cases from January 2015. These smaller units could be placed closer to communities with new infections than the ETC they supplemented.

Incentives to self-reporting–patient feeding, and provision of good medical care for those triaged as Ebola-negative–were adopted on the basis of advice provided by social scientists [3, 5]. The fundamental aims and objective of the CCC was to isolate patients in places where there were no ETC. A news report in the British Medical Journal from the 12th November 2014 [6] summarises the controversy the CCC plan provoked. A representative of *Medecins sans Frontieres* (MSF) went before the UK parliamentary international development committee to argue "the way the CCC are operating, the way they are putting responsibilities on the community, and the way they are designed, is not something MSF is behind at this point". An MSF official with experience in Sierra Leone added that "existing holding centres are close to the patients already". This view was contradicted by an epidemiologist who stated: "we need to have facilities closer to the patients (. . .) transporting patients for hours in the back of uncomfortable ambulances is (. . .) not conducive to patients coming forward to getting early treatment" [6]. After weighing the arguments the authorities in Sierra Leone and UK gave a green light to proceed.

CCC were intended to serve as accessible triage units in areas where numbers of cases were rising. They provided for safe handling of patients waiting for test results, with subsequent transfer to ETC for those with a positive diagnosis, and treatment for those found to have other conditions. This paper documents community responses to CCC, and explains why this development was seen, locally, in a positive light. These new Ebola response centres were viewed with scepticism by some international responders, who feared they would spread infection, but were seen in a more positive light by local communities. It is shown that a major factor was that CCC accommodated local cultural expectations regarding the role of the family in care for the sick. Loved ones, both living and dead, were treated with respect, and other diseases were also treated. Built to a partially open design, CCC allowed families some possibility to monitor a patient's progress [5].

The CCC approach belongs to a broader effort to tackle public health challenges through community mobilization (see for example [7], [8] and [9]) The present paper aims to document and understand community perceptions–positive and negative—regarding the CCC approach, based on qualitative analysis of opinions expressed in a series of community-level focus groups. The aims and objectives of the focus group study were three-fold:

i. To assess community response to CCC, given that it was an untried approach, and there was known hostility at community level to larger more centralized ETC.

ii. To assess whether community responses varied by gender, age or location (with or without a CCC)

iii. To examine responses in relation to a set of key concerns and constraints, four of which (access to the facility, visiting and feeding, burial and land acquisition) are discussed in the paper. Other responses will be discussed elsewhere.

It should be stressed that our aim was specifically to access community-level responses, rather than individual views of e.g. patients or survivors. This is because in rural Sierra Leone, where villages are largely self-managed entities, the community consensus is an important element in determining whether policy interventions succeed. An open focus group approach is the appropriate means to gather information at the community level.

## Materials and methods

The basic design of a CCC is described in [10]. The CCC was typically an 8–10 bed facility in tents (tarpaulin) or a repurposed local building (such as a school), staffed by "volunteers", mostly professionals with medical training, but lacking a Ministry of Health payroll number, and various manual workers, such as guards, cleaners and cooks. Some of the volunteers and most of the manual workers were hired from within the local community, a factor important in gaining trust of patients and their families. All the CCC studied had a water supply, latrines, and security. The layout was divided into "red" and "green" zones [10]. Entry to the "red" zone was barred to all except staff correctly attired in PPE. Some CCC had light at night, supplied by generators. Carers could not attend to suspect EVD patients during the night unless a CCC had electricity [10]. The ICAP study reports that "no sites (visited by the team) were aware of any HCWs (Health Care Workers) who had contracted EVD from their work at the site" [10]. Nursing staff triaged sick persons as soon as they reported. Those without signs of Ebola were treated for malaria, or other diseases and sent home, under observation. Blood samples were taken from those admitted. The aim was to have a laboratory-confirmed result within two days [2, 10]. Confirmed cases of Ebola were transferred by ambulance to an ETC. Some died before diagnosis could be confirmed and were buried by a CCC "safe burial" team.

Data for the present study were collected as part of a formative assessment of the impact of CCC conducted in February 2015 in 14 villages (grouped in seven pairs, one village with a CCC and a second village referring patients to the CCC in the first village) in seven chiefdoms in northern and eastern Sierra Leone. Each referral village belonged to the same chiefdom section as its matching CCC village. A section is the lowest administrative unit in provincial Sierra Leone, typically grouping a handful of villages within a 4–5 km radius.

Choice of the seven chiefdoms was purposive, after consultation with local authorities and responders. Originally, the plan was to choose two chiefdoms in each of four districts (Kambia, Kono, Port Loko and Tonkolili) where Ebola infection chains were still active, but logistical constraints confined the research team to one chiefdom in Kambia District. We also had to bear in mind practical considerations, of reasonable accessibility and a suitable camping site for the team (lodging with villagers–our usual practice—was not allowed under Ebola regulations so the team took tents and did all its own food preparation and housekeeping). CCCs varied somewhat in design and facilities. We did not try to reflect this in our sample design. This is because our focus was on communities, and communities did not experience variations in design. In each case they were comparing their own CCC with what pertained previously–treatment in a village health facility or a distant ETC. It was more important to us to get a good spread of communities, bearing in mind the practical constraints mentioned about doing fieldwork in an active epidemic.

Focus group discussions, lasting typically between one and two hours, were held in all 14 villages. Four focus group meetings (for elders, men, women and youths) per village were held simultaneously to ensure independent responses. First there was an introductory meeting with the village chief and elders. It was explained that all villagers were invited, if they chose to come, but that there would be separate parallel sessions for youth, older women, older men, and elders. It should be added that "youth" has a specific meaning in rural Sierra Leone, referring to a person not yet thought to have seniority in village affairs. The age range is rather vague–generally from 20–35 (i.e. young adult) but it is not unusual to see older people of low social status sort with the youth. We did not include children, thus obviating the issue of obtaining parental consent. Locations for the different groups were announced, and people were told they could choose the group they thought most appropriate to them. Having chosen their groups participants then gave informed consent.

All the various groups were adequately attended. We recorded no names but took information on age groups, gender (for the mixed focus groups for "youth" and "elders") and occupations. These data give some sense of the representativeness of each meeting. There were 56 meetings in all. A total of 1051 people participated and 3399 statements were recorded.

A single question was used to start discussion: what (good or bad) changes have there been in your community in the last year? In all groups the topic of Ebola was quickly reached. Facilitators were supplied with a list of topic prompts to guide discussion further. In some cases, topic prompts were used sparingly because there was a natural flow to the discussion. Speakers were guaranteed anonymity as part of an informed consent procedure.

A card system was used to keep account of the type of speaker, when they joined the conversation, and how many times they spoke, without having to record names. Two sequences of numbered cards known as "run order" (labelling respondents as A, B, C, etc.) and "speaking order" cards (numbering the times each respondent spoke–A1 A2, A3, etc.) were distributed and cashed in each time a participant raised a hand to speak. Run order and speaking order details were attached to statements as facilitators wrote them down.

The card tracking system allows the analyst to discover patterns of responses–e.g. whether certain opinions were favoured in some but not all of the four groups, whether certain people dominated the conversation, or whether the expression of a particular opinion by a person of higher status or greater seniority was confirmed by echoing statements from others who spoke later in a sequence. There is insufficient space in the present paper to offer the fine-grained analysis made possible by this tracking system, but the data are supplied on line, and we plan further analysis. For present descriptive purposes, a few summary numbers are supplied below to give an indication of the importance of particular topics to different strata within the sample.

Additionally, each group made its own house rules (e.g. to speak in a moderate voice) and to encourage as many persons as possible to contribute to the discussions.

Each focus group was run by two facilitators. Facilitator One led the discussion, asking a start-up question about diseases affecting the community. The facilitator confirmed that groups could talk about Ebola response once discussants had first raised it, and specifically about the CCC, as they wished. The prompt list was used to ensure a degree of consistency across groups. Facilitator Two managed, monitored and took notes of body language to assess reactions (in the first place to guard against distress), ran the card tracking system and wrote down and translated the discussion.

The 3399 recorded statements were grouped into twelve broad themes (see comment on aims and objectives above). Statements were then classed as descriptive (type-1) or evaluative (type-2). A statement would be classed as descriptive if it simply stated a fact–for example "the CCC had a generator". It would be considered evaluative if an opinion was expressed–for example, that "CCC staff showed sympathy for the patient". This resulted in 1367 (40%) type-1 and 2032 (60%) type-2 statements. Evaluative statements were the focus of our analysis.

Due to length constraints results for only four of the twelve themes are presented in this paper, covering about a third of the total data (Table 1). These four topics have been chosen to

**Table 1. Overview of the data subset.**

| Topic | Speakers | Type-2 Statements |
|---|---|---|
| Distance | 89 | 96 |
| Visits | 195 | 227 |
| Burial | 147 | 150 |
| Land | 131 | 144 |
| **Total** | **562** | **617** |

reflect the top priorities of the focus group participants, in terms of evaluative statements. The four themes are i) access to Ebola treatment facilities, ii) visiting and feeding patients, iii) burial, funeral ceremonies, and reporting death of patients, and iv) acquiring land to set up a CCC.

Type 2 statements can be viewed from the perspective of framing assumptions derived from Mary Douglas' theory of social ordering [11, 12, 13, 14]. Douglas recognizes four forms of social ordering–isolate, hierarchical, enclave and individualistic ordering–derived from two universal dimensions of social life (social integration and social regulation). Enclave and hierarchical ordering are of particular relevance to Ebola response in Sierra Leone. Villages in Sierra Leone operate as political enclaves [15]. They are largely self-governing. For example, a survey of village dispute resolution [16] showed that only 4 per cent of disputes were settled in (government-supervised) local courts—96 per cent of cases involved reference to family heads or other trusted elders.

The decision-making process follows the patterns of village social structure. By contrast, a large part of the Ebola response involved hierarchical ordering. An example would be the front-line medical staff such as nurses and Community Health Officers under the direct command of District Medical Officers and senior officials of the Ministry of Health and Sanitation. Our focus group data are the "enclave" portion of a fuller data set to be analysed elsewhere. Here, results are presented descriptively.

## Ethics statement

The data were gathered as part of an independent review of CCCs undertaken by a team recruited by the Institute of Development Studies at University of Sussex at the height of the Ebola crisis in in Sierra Leone in late December 2015. The urgent objective was to assess whether the new policy of building CCCs had any major flaws when viewed from the perspective of communities. The work was considered to be "impact assessment" and not primary research. The research protocol for community focus group discussions had been previously developed by team members and approved by ethical review boards in Njala and Wageningen universities.

The team was also required to apply institutional ethical guidelines. These ensured that all participation by villagers was voluntary, that data collection was undertaken under a protocol guaranteeing participant confidentiality, and that community leaders gave consent for the holding of consultative focus groups. All human subjects were adult. Informed consent was oral because only a minority of participants could read and write. The process involved the reading out of a statement of informed consent after which participants took time to reach collective agreement. This was reported to the Paramount Chief, who served as custodian of the community interest. No patient samples or experimental procedures were involved.

## Results

### Access to Ebola treatment facilities

Of the evaluative comments grouped under this theme, it was found that 74 statements (77%) directly referred to expectations concerning distance and family/inter-family involvement in care for the sick, 47 from males, and 27 from females. The distance of ETC was mentioned 35 times. Statements expressed specific obstacles such as the cost of transport, the hazard of a long journey for a seriously sick patient, and the difficulties families faced in maintaining contact with the patient in a distant location.

Many of these comments came from the four sample villages in Kono and Tonkolili districts. Ebola cases in Kono were at first directed to the ETC in Kenema, a distance of about 100

km., but all of it over very poor roads. Patients from the two Tonkolili villages had to travel to the ETC in Bo, more than 150 km., and later to Makeni, a distance of about 80 km. The problem with both journeys was the first part on rutted tracks.

The advantages of a local Ebola treatment facility (CCC) were mentioned 39 times. Reasons included the ability to maintain contact with the patient, and opportunities to fulfil expected duties of care. Local values are specifically evident in the following comment on the community's role in the decision of a sick person to report for diagnosis. As one respondent put it: "We the community members monitor each other's health issues and can easily advise anyone sick to go to the CCC."

Exclusion from the group is one of the most severe social sanctions that the enclaved community possesses [15]. It was important that patients did not feel abandoned by their families, even if visitors could only gather at the margins of the "red zone" and converse at a distance. This, of course, was more feasible in the small-scale CCC than in the much larger, highly secure ETC.

## Visiting and feeding patients

Visiting and food sharing is an important way in which enclaved community social bonds are expressed in the Sierra Leone countryside. Villagers are committed to a lifetime of visits with those to whom they are linked by kinship, marriage or patronage [15, 17]. Such visits cover a wide variety of social reasons. Sick visiting is always a high priority, especially for family members and in-laws. A visit to the sick involves offering prayers and good wishes, consoling and encouraging the sick person, and giving a helping hand to the carers. Food is often brought and shared.

Ebola disrupted normal patterns of sick visiting, and this threatened the expression of community solidarities. Initially, communities resisted the changes that were required. Patients were sometimes hidden, and burials were carried out in secret. But the disease is very dramatic, and quickly reveals, through the way it spreads from the first victim to close family carers, that it is spread by direct bodily contact. Faced with the losing a family member to a distant ETC or attempting home care, villagers experimented with ways of protecting themselves, while continuing to care for victims of the disease. Evidence concerning the use of improvised protective measures, such as plastic bags to cover the hands and face when nursing patients has been reported [1].

Families also continued to emphasise the importance of home feeding as necessary to recovery. Any such help was impossible in a distant ETC, but it became possible in a local CCC, where many of the kitchen staff were recruited from the village, and willing to accommodate the wishes of villagers who brought home-cooked food for Ebola victims. ETC became better as time went by at community liaison [18] but distance ruled out home-cooked food.

In all, 227 statements by 195 people were made in response to prompts about whether the sick could be visited in CCC, and under what conditions; all referral villages were located in the same chiefdom administrative section as the village within which the CCC was located, so people in these villages were also asked what they knew and felt about the CCC, even though it was not located in their village. A substantial proportion (56%) of all responses concerned whether or not families were permitted to visit and help care for patients in the CCC.

Statements were often carefully qualified–for example, that centres allowed families to visit and communicate with patients, but not to enter "red zones", or that home food was accepted, but families could not, themselves, serve it to patients, etc. About half of all discussants insisted that family visits and care were not permitted or encouraged. Discussants from villages with CCC were more likely to state that there was a possibility to visit patients, although this was also mentioned frequently in statements from the referral villages. A smaller number of

responses commented that CCC provided free treatment, treatment for other diseases, and rapid testing for EVD. Feeding for patients was mentioned in ten per cent of statements. CCC care in non-Ebola cases was also sometimes highlighted. One man reported that "my woman had a severe stomach ache, and she was treated, and given food at the centre, free of charge."

## Burial, funeral ceremonies, and reporting death of patients

Focus groups often raised issues relating to the safe burial regulations introduced to break Ebola infection chains. Official procedures required that corpses were routinely swabbed to assess whether the deceased had died of EVD. From August 2014 all burials had to be carried out by a trained "safe burial" team, whether the swab was positive or not. The team would spray the corpse with chlorine and place it in a body bag. It would then be buried in a hastily prepared grave with only a minimum of ceremony. Initially, the family was excluded, but from November 2014 families were allowed to participate at a distance. All contact with the body was forbidden.

Burial teams also operated from some CCC. But here it was more feasible to notify families, and to arrange burial in the victim's own community, since this was now near at hand. Families were allowed to attend burials and observe at a distance. But repeated calls by communities to provide volunteers to be given the training and protective equipment to carry out their own safe burials were ignored or rejected by the international response.

Given the importance of funerals as ways of cementing social relations in enclave-ordered communities it was expected that many group comments would focus on the importance of involvement of families in burial. But since "safe burial" during the Ebola crisis involved new regulations imposed by the state it was also expected that some comments might reflect the hierarchical ordering under which village chiefs and elders administer rural Sierra Leone 's system of "customary" local government.

These expectations were met. In all there were 150 comments from focus groups pertaining to burial, funeral ceremonies, and reporting the death of Ebola patients. Of the evaluative statements, 71 (47%) were classed as being aligned with enclave-ordered perspectives and 33 (22%) were classed as being aligned with hierarchically ordered perspectives.

Instances of "enclave" perspectives included demands or suggestions that families be trained or empowered to carry out funerals. "We will wear protective gear and do the burial ourselves" was one statement. Another speaker insisted: "Let the CCC give [us] protective gear (gloves, and PPE) and hand over the corpse to the family members, who will wash and dress [it] and pray on the corpse."People were not opposed to protective measures as such, but wanted family members to be trained to apply these measures: "The CCC [staff] should bring the corpse to the family and give the family protective gear to bury their dead."

Other comments requested burial teams to permit family members to attend burials, wanted teams to bury victims on family land or in the victim's village, and hoped that "safe burials" by CCC staff would follow village ritual practices. The idea of excluding families from burials was a source of concern; one commentator remarked: "the government will bury them; the family will never see the corpse", implying that a "government" burial would be a scandal.

A second, less extensive set of 34 statements, contained items reflecting or endorsing the government-mandated Ebola bye-laws. Based on rules first developed by chiefs in Kailahun District (the epicentre of the disease in Sierra Leone) these requirements were promulgated as a national set of bye-laws for Ebola control in August 2014 [19]. Typical statements repeat bye-laws or refer to epidemiological issues. For example: "The burial team will bury the way authorities (require), (supervised) by health officers" or "let the burial team continue to do the burial, as they have been doing" and "I will advise (that] we call the burial team to come and do the burial, to avoid the spread of the sickness."

Thirdly, enclave ordering imposes a strong emphasis on the manner of reporting death. It requires it to be done in a timely but formal manner, by those with direct knowledge of the circumstances, reporting to heads of the affected families. Anything casual, approaching rumour or gossip, is frowned upon. In the case of an elder, a word out of place may attract a fine. This is because the enclaved rural community in Sierra Leone is a self-monitoring entity. Families must inform each other; reliance on state machinery for reporting births and deaths is not yet accepted as a matter of course.

Focus group members were asked to discuss their preferred ways to be informed about the deaths of Ebola victims. As expected, many comments stressed the importance of face-to-face reports from the case handling centre to the appropriate family head. It was not clear from focus group comments whether there was an agreed protocol for reporting deaths to families. It was said that sometimes reporting went via chiefs. But it was seen as helpful that centres were close enough to permit visits, and some deaths were reported in the required face-to-face manner, perhaps because they employed local people, who knew the families and what to do. In fact, tradition is flexible, and a good number of people reported that they considered a phone call or radio announcement to be acceptable. These are widely used media in funeral practice in Sierra Leone. Such announcements allow scattered family members to be fully and quickly informed. The issue is more about the routing and timing than the medium. The key feature is that the message goes in a timely manner to the correct recipient.

## Acquiring land to set up a CCC

A potentially troublesome clash of institutional values between communities and responders concerned the acquisition of sites for CCC. Land is a controversial issue in rural Sierra Leone because it brings up a compromise made by the British colonial power at the beginning of the 20th century over the authority of the state versus land-owning families. The Paramount Chief is "custodian" of the land but brokers the competing interests of families and government. International Ebola responders wanted land for CCC quickly. For this they turned to Paramount Chiefs for rapid action, but there was often a push-back from families, who pointed out that the ultimate decision rested with them. CCC were welcome, but not necessarily on "our" land. In this respect, there was a clash of interest between responders and communities.

Names of landowning families are well known to the communities though people tend not to advertise ownership openly. In any land decision both landowning families and chiefs must be involved. A participant in one focus group discussion put the point neatly, when stating that "The chief should be approached for him to lead you to the landowner. The land-owner will now negotiate with the person who wants the land."

The focus group data show that families offered land free as a gesture of community solidarity, but welcomed or required acknowledgment: "We were consulted by the Chief and we accepted to give the land even though they did not pay for it, but we were respected in the process of gaining the land." This demand for respect served to reinforce the basic local viewpoint that all land is family land, and cannot be expropriated, even in an emergency. Family sovereignty is also apparent in focus group extracts expressing disgruntlement and compromise over land (see supporting data). Some statements expressed both dissatisfaction and concern: "We were not happy (with) how we were treated. We had wanted to cause confusion [create trouble] but [we did not because] we were thinking of the Ebola disease."

## Discussion

The present study has analysed family responses to Ebola community care centres. Some of the ways CCC opened pathways to community participation in Ebola care, e.g. through family

involvement in food preparation and in inclusion in burial processes, have been traced. The evidence suggests that CCC were well received by communities and led to improved relationships of trust between communities and responders, despite some problems in set-up and execution.

Location of case-handling facilities proved to be a crucial issue. Some responders felt that in a small country with good main roads, accessibility to case-handling centres was not a major problem [6]. This was to ignore the poor state of many access roads, and to misunderstand the obligations placed upon family members to be present in helping care for the sick. Accessibility is not to be measured in miles by ambulance but in terms of the logistical challenges associated the family accompanying the patient. For example, people in Kambia were reluctant to allow their loved ones to be taken to the ETC in Port Loko, only twenty miles away on a very good road, because they did not have the connections and resources needed for family attendance. Who would prepare food and be on hand when the patient needed encouragement?

CCC helped address this issue by bringing case-handling closer to families. Previous work has offered evidence of shorter times to presentation for CCCs relative to ETCs [2]. When the only option was referral to an ETC, families hid patients with high fevers, but once the CCC option was available families were more forthcoming in bringing cases for assessment. Most diagnoses were of malaria, and this was treated, and the patient discharged, to the relief of the family. If the diagnosis was of EVD, the CCC helped cushion the shock for both patient and families. Where before there was panic, and an ambulance driving to Bo or Kenema at high speed with sirens wailing, there was now a more calm and considerate process. It would be explained to the family that the best chance of survival was transfer to an ETC. But the CCC was equipped to accommodate an EVD patient if it was too late to arrange safe transport. The carers at the CCC were often themselves members of the local community, and their advice was trusted. CCC were small enough, and the structures were physically open enough, to allow family members to communicate directly with the patients from the perimeter of the facility. Messages would be sent to patients to hang in there, and not to lose heart; community expectations for family support were audibly maintained.

Directly caring for the sick and sharing of family food are important ways in which families reinforce social solidarity. Focus group discussants insisted that this expression of solidarity helps patients survive a devastating disease. Home cooking encourages the will to live. They view it as an essential part of treatment. This insistence provides, in turn, some lessons for the improvement of ETC, modified to function more like CCC in social terms. For instance, transport could be hired to allow family members to follow referrals. Camps could have been built for their accommodation next to each ETC. Equipping such camps with kitchens stocked with firewood, water and other supplies to facilitate preparation of familiar food would be no more complex than building and equipping the kitchens already part of standard ETC design. Members of families would then be able to continue to take part in the monitoring and feeding of the patient, even if visiting the "red zone" remained out of bounds.

The focus group material brought out the enormous significance of the issue of safe and dignified burial. Some patients died in CCC, either from non-EVD diseases or because it was not possible to transfer them to an ETC. CCC were built at a time when the problem of safe and dignified burial had been recognised by the authorities. Although "safe burial" crews did the actual internments staff encouraged families to attend, and this was appreciated by discussants. Attendance was feasible because the families lived locally. This threw into contrast a major problem with the ETC, that family members were often many miles distant with poor communications and did not know when their loved ones had died or where they were buried.

Discussants were divided about the role of the family in Ebola burial. Some accepted burial by trained teams as necessity, both for biosafety and in respect of national byelaws on "safe

burial". Others argued strongly that families could and should have been equipped and trained to do safe burial for themselves, because it was said (for example) that burial teams were never on time, that corpses were not washed and dressed in *kasankei* (grave clothes), that there was no final farewell and prayers for the dead, and that burial team members were strangers to the deceased.

The issue of the land acquisition for building CCC proved somewhat controversial. Land belongs to families, and not to government or the chiefs. Focus group materials evidenced difference between those who believed the government or chiefs had a right to acquire land to build CCC and a greater number of discussants who insisted that the land belonged to landowning families, who should have been consulted, despite the urgency of the epidemic. Examples of "good practice"–CCC going through the right channels, for example—were also noted. Some families offered land specifically to help communities fight EVD. But disgruntlements over land sometimes surfaced, even though generally set on one side because of the epidemic emergency. Given the small amount of land needed for both CCC and ETC it is perhaps surprising that acquisitions of temporary leases proved so contentious. The more general point needs to be taken that an improper approach to land acquisition is seen as a threat to community cohesion.

It is perhaps worth adding that there were few obvious differences between the opinions expressed in the CCC centre villages and the referral villages, other than some comments about whether the right village had been chosen–some would have preferred the referral village to have been the CCC village, others were glad it was not.

The CCC intervention helped draw attention to the role of local knowledge brokers in the process of community adaptation to the risks of Ebola. The shift to a policy of "safe and dignified burial" in November 2014 brought out the role played by Imams and Pastors in gaining acceptability for changes in burial practice. CCC later benefited by being able to make use and influence of these local knowledge brokers.

Herbalists and Traditional Birth Assistants (TBAs) are equally influential in rural communities. They are trusted in villages because they are resident, and accessible when needed. During the EVD epidemic in the Sierra Leone they were deliberately by-passed. This was because of a fear that incautious handling of Ebola patients would spread infection. Early on in the epidemic stories circulated about "witch doctors" pretending to cure Ebola but instead spreading the disease. The government then banned herbalists from practising for the duration of the epidemic. From the perspective of communities this might have been a mistake. Herbalists spread the disease in a few early instances because they did not at that stage know what they were dealing with.

They became agents of infection not through wilfulness, but because they were the helpers of last resort. Professionally trained medical practitioners also spread the disease in the earliest stages of the epidemic, when it remained unidentified, because they lacked the training and resources to deal with it. Like doctors and nurses, traditional birth attendants and herbalists quickly learned about the dangers of EVD and modified their practices. As highly respected authorities, they could have been used to spread correct knowledge of the disease. This is a topic on which further research is now needed in Sierra Leone (for Ghana, see [20]).

Home care is another salient point for debate. There is some evidence that CCC helped to get people to report EVD cases earlier in Sierra Leone, and so contributed to epidemic downturn [2]. But some communities were still too remote for patients to be moved quickly. In such a case a hammock might be needed. This is an expensive process and takes time to arrange. Moving the patient in the "wet" phase could be highly hazardous to the carriers. Guidance and supplies to permit safer home care are potentially helpful for such extreme cases.

A protocol for coping with an Ebola patient at home while waiting for help was released by the US Centers for Disease Control in November 2014 [1]. Some knowledge about quarantining patients in farm huts was retained by communities from an era in which smallpox was still a scourge. A basic rule of a single carer, and the rest of the household providing distant backup seems to have been applied. Some of this old knowledge was carried over to Ebola. It is perhaps not insignificant that the Mende word for Ebola–*bondawote*–means "family turn away"

In sum, then, the policy of offering care for Ebola victims in small, quickly constructed handling units placed where EVD case numbers were rising, to complement large-scale ETC, received largely positive endorsement from rural communities in Sierra Leone. This applied both to CCC locations and to neighbouring communities. Evaluation, accessed through focus group discussions, confirms that CCC were compatible with community values concerning access to and family care for the sick. "Safe burial" was more controversial. This directly challenged a ritual activity seen as vital to maintaining good relations within and between rural families. Focus groups also found land acquisition to build CCC a controversial topic, but this can be interpreted as reflecting a larger problem of relations between communities and central government unresolved since the colonial era. This was not an institutional clash specifically related to EVD.

It is not advocated that CCC should replace the ETC in future Ebola outbreaks, such as that currently threatening parts the Democratic Republic of Congo. The main conclusion of this study is that evidence for the social acceptability of CCC in rural communities in Sierra Leone reinforces the case for a combined strategy, in which CCC are deployed as triage centres to screen out and treat malaria and other diseases, while directing EVD cases towards further specialist care in ETC. CCC also serve as effective learning sites through which communities can come to terms with the biological challenges of EVD without local norms of community support in sickness being undermined. In this respect, the experience of localised case handling in Sierra Leone offers lessons that can be usefully disseminated throughout the wider field of Ebola response.

## Supporting information

**S1 File. Statements sorted.**
(XLSX)

## Acknowledgments

The authors wish to wholeheartedly thank all the people in Port Loko, Kambia, Tonkolili and Kono Districts who contributed to this report, including the Paramount Chiefs, Village Chiefs, the elderly, and community and advocacy groups for sharing their insights and time.

The research assistants brave enough to carry out fieldwork at the peak of the Ebola epidemic, sleeping in open court barries, and determined under trying circumstances to listen and learn from people in the communities were Vandy Kanneh, Idrissa Sesay, Philip Musa Lahai, Ramatu Samawoh, Francis Baigeh Johnson, Famata Binta Jalloh, Daniel Mokuwa and Sao Bockarie. A special appreciations and thanks goes to them all. We are very grateful to Paul Richards for his enduring support and input to the paper and underlying research. Many thanks also to the reviewers for their critical comments and helpful suggestions to improve the paper.

We would like to thank members of the Ebola Response Anthropology Platform (ERAP) for their guidance and support in designing the data collection.

This paper draws on material originally collected by the first author and her team as part of an impact assessment of CCC in February 2015 organised by the Institute of Development Studies at the University of Sussex (IDS). The permission of IDS to draw on the data for the purposes of this paper is gratefully acknowledged. We thank IDS, the Development Economics group of Wageningen University and the Dutch Council for Scientific research (NWO) for institutional support.

## Author Contributions

**Data curation:** Esther Yei Mokuwa.

**Formal analysis:** Esther Yei Mokuwa, Harro Maat.

**Investigation:** Esther Yei Mokuwa.

**Methodology:** Esther Yei Mokuwa, Harro Maat.

**Supervision:** Harro Maat.

**Validation:** Esther Yei Mokuwa.

**Writing – original draft:** Esther Yei Mokuwa.

**Writing – review & editing:** Harro Maat.

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
