## [Decision Letter · Decision Letter 0]

14 Oct 2019

Dear Dr Maat:

Thank you very much for submitting your manuscript "Rural populations exposed to Ebola Virus Disease respond positively to localised case handling: evidence from Sierra Leone" (#PNTD-D-19-01217) for review by PLOS Neglected Tropical Diseases. Your manuscript was fully evaluated at the editorial level and by independent peer reviewers. The reviewers appreciated the attention to an important problem, but raised some substantial concerns about the manuscript as it currently stands. These issues must be addressed before we would be willing to consider a revised version of your study. We cannot, of course, promise publication at that time.

We therefore ask you to modify the manuscript according to the review recommendations before we can consider your manuscript for acceptance. Your revisions should address the specific points made by each reviewer. 

When you are ready to resubmit, please be prepared to upload the following:

(1) A letter containing a detailed list of your responses to the review comments and a description of the changes you have made in the manuscript.

(2) Two versions of the manuscript: one with either highlights or tracked changes denoting where the text has been changed (uploaded as a "Revised Article with Changes Highlighted" file); the other a clean version (uploaded as the article file).

(3) If available, a striking still image (a new image if one is available or an existing one from within your manuscript). If your manuscript is accepted for publication, this image may be featured on our website. Images should ideally be high resolution, eye-catching, single panel images; where one is available, please use 'add file' at the time of resubmission and select 'striking image' as the file type. 

Please provide a short caption, including credits, uploaded as a separate "Other" file. If your image is from someone other than yourself, please ensure that the artist has read and agreed to the terms and conditions of the Creative Commons Attribution License at http://journals.plos.org/plosntds/s/content-license (NOTE: we cannot publish copyrighted images). 

(4) If applicable, we encourage you to add a list of accession numbers/ID numbers for genes and proteins mentioned in the text (these should be listed as a paragraph at the end of the manuscript). You can supply accession numbers for any database, so long as the database is publicly accessible and stable. Examples include LocusLink and SwissProt.

(5) To enhance the reproducibility of your results, we recommend that you deposit your laboratory protocols in protocols.io, where a protocol can be assigned its own identifier (DOI) such that it can be cited independently in the future. For instructions see http://journals.plos.org/plosntds/s/submission-guidelines#loc-methods

While revising your submission, please upload your figure files to the Preflight Analysis and Conversion Engine (PACE) digital diagnostic tool, https://pacev2.apexcovantage.com/ PACE helps ensure that figures meet PLOS requirements. To use PACE, you must first register as a user. Then, login and navigate to the UPLOAD tab, where you will find detailed instructions on how to use the tool. If you encounter any issues or have any questions when using PACE, please email us at figures@plos.org.

We hope to receive your revised manuscript by Dec 13 2019 11:59PM. If you anticipate any delay in its return, we ask that you let us know the expected resubmission date by replying to this email.

To submit a revision, go to https://www.editorialmanager.com/pntd/ and log in as an Author. You will see a menu item call Submission Needing Revision. You will find your submission record there. 

Sincerely,

Eric Mossel, Ph.D.

Guest Editor

Samuel Scarpino

Deputy Editor

Two of the reviewers raise substantial concerns regarding the use of human subjects in this study. In addition to the other comments and suggestions, it is critical that these concerns be thoroughly addressed before a final decision on publication can be made.

Reviewer's Responses to Questions

**Key Review Criteria Required for Acceptance?**

**Methods**

-Are the objectives of the study clearly articulated with a clear testable hypothesis stated?

-Is the study design appropriate to address the stated objectives?

-Is the population clearly described and appropriate for the hypothesis being tested?

-Is the sample size sufficient to ensure adequate power to address the hypothesis being tested?

-Were correct statistical analysis used to support conclusions?

-Are there concerns about ethical or regulatory requirements being met?

Reviewer #1: Per comments

Reviewer #2: Methods

- Aim of the study should be more clearly articulated – at the moment the authors have stated that these data were collected as ‘part of an assessment of the impact of CCC’ – could they provide more specific aims/objectives of this study and how it differs from the overall assessment of which this is a part?

- Did all villagers participate? Unclear how participants were identified. This section would benefit from some additional detail.

- Topic guides – were these semi-structured/unstructured? Who conducted the fieldwork? What level of training did they receive?

- If the FGDs were conducted simultaneously, and each had two facilitators, does this mean the field team was comprised of 8 people?

- What is meant by a ‘statement’ in the context of specifying the number identified?

- What is meant by ‘descriptive’ vs ‘evaluative’ statement types?

- What analytical approach was utilised? How were themes developed and determined?

- Line 162-163 – unclear what differences (between villagers and health agencies) are explored and how the views of agencies are represented given that the data collection approach described here only focuses on interviews with community members. More detail on methods required. 

- Table 1 – not clear what this depicts. 

- Lines 169-181 – did the Douglas schematic serve as an analytical framework for the data? Make clearer. Why this approach rather than another?

Ethics

- What steps were taken to manage vulnerability or distress?

- Could the collective approach to agreeing participation have risked that some participants felt compelled to participate? What steps were taken to avoid this?

Reviewer #3: The methods seem clear however somewhat confusing.

the statement that "We have chosen these four topics as they reveal differences between villagers and (inter)national health agencies most prominently" -- this is not clearly explained

Also unclear the time between CCC set up and data collection (feb 2015 stated, but not clear what the gap between set up was)

explanation of CCCs remains very limited - as if there is only one type of CCC, when in fact there were multiple models (several reports in fact simply explaining the multitude of types and variations) - unclear how this was accounted for. It could at least be mentioned as something that wasnt specified or looked at (but should be acknowledged)

the SDB aspect is a bit confusing - the analysis is on CCCs. the topics (four) that were the point of this paper were reportedly because they most frequently arose in discussions. However, on line 284 it states "Focus groups were asked to discuss the impact of burial regulations introduced to break Ebola infection chains" -- i'm then not clear on the link with CCCs. Unless the link was different aspects required to break transmission? (ex. SDBs, CCCs etc). I think the SDB aspect is not so clear in terms of how this comes out/ is linked to CCCs. Following this, the analysis is better explained, however I feel that this should be better explained earlier / later on (ex. 323-325 is not related to CCCs)

As well, the paper implies that all CCCs were supporting SDBs, which I do not believe was the case.

**Results**

-Does the analysis presented match the analysis plan?

-Are the results clearly and completely presented?

-Are the figures (Tables, Images) of sufficient quality for clarity?

Reviewer #1: Per comments

Reviewer #2: Results

- Line 207 “all CCC had a water supply etc…” – all CCC in the country or all the CCCs that were the focus of the research? If the former, please add references. CCCs visited by this reviewer were not fully equipped, although that might have been at an earlier stage in the outbreak. 

- Very unclear how this distinction between descriptive and evaluative statements was determined. What was the analytical process here? 

- What is the significance of drawing this distinction?

- Lines 222-227: what is the significance of the quantitative breakdown? E.g. male vs female – how does this relate to the gender split among participants? Did men express this sentiment more because it was more important to them (and if so, why?), or does this just reflect the demographics within the sample?

- Unless there is a very good reason for presenting qualitative data quantitatively, it is generally not good practice. In this article it is not clear why this approach has been taken, or indeed how to interpret the quantitative data presented. What is the significance of the number of times a particular issue was raised? What is the base here? How does it relate to the overall sample? 

- The quotation used at line 233-234 doesn’t seem to relate to the point being made directly above insofar as it seems to focus more on community referral than on providing care to patients inside a CCC. 

- As the outbreak progressed, ETCs changed their practice and found ways to provide opportunities for interaction between patients and family members. The authors should acknowledge that at lines 236-239, and provide references.

- Line 244: What is meant by “villagers are committed to a lifetime of visits”?

- Lines 243-248 – refs needed to support descriptions of care practice in rural SL.

- How do the authors interpret the fact that participants from CCC and control villages seemed to think that there was more chance of family visits at the CCC, despite the fact that control villagers would not have had experience of a CCC? 

- Lines 295-297: unclear what the significance is of explaining what was expected from FGDs unless clear data are shown to illustrate how these expectations were confounded. 

- Lines 297-300: it would be interesting if the authors explored in more depth the conflict between the two types of social ordering they identify as critical in the SL Ebola period.

- Were there differences, across any of the themes, depending on whether data collection took place in a control or CCC village? If not, what do the authors posit as reasons for that? If the overall hypothesis is that the CCC was a more acceptable intervention then you might expect to see differences in the way they were discussed. 

- Line 309: unclear what is meant here.

- Line 316-317: in what way do these words ‘hint at a stand-off’?

- The authors over-rely on the distinction between hierarchical and enclave social ordering as a means of theoretically grounding their findings. How much evidence is there that rural Sierra Leonean villages are ordered as ‘enclaves’, and is this sufficient as a means to describe all the phenomena identified during this research? Might the experience of an emergency have affected local ways of viewing and constructing ordering? Is there evidence from other similar non-emergent settings or emergencies that the authors could discuss to explore this further?

- Lines 327-333 is there necessarily a tension between EVD death reporting requirements and the traditional systems? What would prevent families from reporting in the traditional way while also alerting the appropriate authorities? 

- Lines 341-345 – if phones or radio announcements are ‘widely used’ in SL, why was it a surprise that these were reported as acceptable ways of receiving the news of the loss of a loved one?

- When reporting results that are necessarily embedded in policy or implementation changes, it would be helpful if the authors could make it clearer when events being reported are based on literature or an events timeline, or based on participant reports e.g. line 364 “some families offered land as a gesture of community solidarity” – is this based on other evidence or does it emanate from the data collected for this study? This point stands throughout much of the results section e.g. for more clarity on which evidence source supports a given conclusion.

Reviewer #3: unclear about the burials in the introduction

this doesnt come up as one of the 4 themes, but appears in summary and introduction.

**Conclusions**

-Are the conclusions supported by the data presented?

-Are the limitations of analysis clearly described?

-Do the authors discuss how these data can be helpful to advance our understanding of the topic under study?

-Is public health relevance addressed?

Reviewer #1: Per comments

Reviewer #2: Discussion

- Lines 385-389: is this example from the data collected for this study or from other evidence? Not clear and needs a quote or references.

- We know that IPC practice at some CCCs was poor, and that staff there struggled with poor supplies, not receiving payroll or hazard pay, and were often dependent on handouts from the community to provide food and other basics – the authors should acknowledge this and some of the other challenges associated with the CCC approach, even if the overall conclusion is that they were a positive intervention from the community engagement perspective.

- Lines 418-420: again, there were challenges with CCCs and to my knowledge there is published evidence suggesting that nosocomial transmission might have been exacerbated at these centres. The authors should make this clear and provide references.

- Lines 421 and elsewhere: the discussion of community perceptions of burial processes should acknowledge that the burials intervention changed throughout SL over the course of the outbreak and moved towards a ‘safe and dignified’ model in which families were invited to observe and attend. Presenting burials from CCCs as the only location where this approach was utilised is misleading. Perhaps the authors could include a timeline showing policy developments for key interventions across the outbreak and explore the specific phenomena emanating from their data collection in light of the wider context, rather than in isolation?

- Lines 443-445 – the evidence presented does not suggest that land acquisitions were as contentious an issue as suggested. Could the authors add some illustrative quotes and explore this further in the associated results section?

- Lines 447-465: this is the first time knowledge brokering is mentioned. If the authors think this is a critical theme (and other published evidence does indeed suggest it might be), could they explore this further in the results section before introducing it in the discussion? Again, some references from other literature would help strengthen the argument about the importance of e.g. traditional healers as knowledge brokers in SL.

- Line 468: This is highly contested and definitely needs supportive referencing. There are many factors that might have affected earlier reporting, and highlighting CCCs as a critical factor should be done with caution and evidentiary support.

Reviewer #3: conclusion and discussion is the strongest component however doesnt link to the complicated methodology and coding/ analysis. 

is there a possibility to reduce the complexity of analysis? the scales seem less pertinent in the discussion/ conclusion or at least not linked. 

however this section is the most interesting / useful.

**Editorial and Data Presentation Modifications?**

Reviewer #1: Per comments

Reviewer #2: (No Response)

Reviewer #3: minor revisions required

editing overall is required

clarity regarding methodology and linking with discussions

clarity around the CCC explanation (even in footnote/ annex) -- consider this paper http://www.ebola-anthropology.net/wp-content/uploads/2015/07/Community-Based-Ebola-Care-Centres_A-Formative-Evaluation1.pdf

**Summary and General Comments**

Reviewer #1: Nice paper - consider publishing with minor revisions

Reviewer #2: Overall, use of evidence needs to be significantly strengthened by:

- Improving referencing throughout

- Making it clearer what evidence has led to a particular conclusion or set of conclusions

- Increasing the use of illustrative comments and elaborating further on their value and contribution to the weight of the evidence e.g. the authors refer to a FGD statements that express disgruntlement but do not present those data or contextualise them. 

- Either removing a quantitative approach to the presentation of qualitative data or explaining in more depth what the significance is of any quantitative conclusions might be.

Reviewer #3: Some sentences are not clear

(example "Nursing staff triaged sick persons when they reported."  incomplete sentence)

similarly "Others – due to lack of availability of ambulances – were cared for in the CCC and discharged if they survived" -- which "others" - also unclear.

line 339 - similar sentence which is unclear "CCC may not have been an agreed policy"

needs some general revision for flow

the inclusion of burials in the summary and introduction is not as clear as the rest of the results shared in intro/summary and should be better explained.

would be useful to have some recmommendations for HOW these conclusions could be considered and what would be challening/ not generally application. there is no additional discussion of considerations for culture/ leaders/ community roles in terms of recommendations/ considerations in current/ future outbreaks

PLOS authors have the option to publish the peer review history of their article (what does this mean?). If published, this will include your full peer review and any attached files.

Reviewer #1: Yes: Paul Pronyk

Reviewer #2: No

Reviewer #3: Yes: Simone E. Carter

---

## [Decision Letter · Decision Letter 1]

4 Jan 2020

Dear Dr Maat,

We are pleased to inform you that your manuscript, "Rural populations exposed to Ebola Virus Disease respond positively to localised case handling: evidence from Sierra Leone", has been editorially accepted for publication at PLOS Neglected Tropical Diseases.

Before your manuscript can be formally accepted and sent to production you will need to complete our formatting changes, which you will receive in a follow up email. Please note: your manuscript will not be scheduled for publication until you have made the required changes.

IMPORTANT NOTES

* Copyediting and Author Proofs: To ensure prompt publication, your manuscript will NOT be subject to detailed copyediting and you will NOT receive a typeset proof for review. The corresponding author will have one final opportunity to correct any errors when sent the requests mentioned above. Please review this version of your manuscript for any errors.

* If you or your institution will be preparing press materials for this manuscript, please inform our press team in advance at plosntds@plos.org. If you need to know your paper's publication date for media purposes, you must coordinate with our press team, and your manuscript will remain under a strict press embargo until the publication date and time. PLOS NTDs may choose to issue a press release for your article. If there is anything that the journal should know, please get in touch.

*Now that your manuscript has been provisionally accepted, please log into EM and update your profile. Go to http://www.editorialmanager.com/pntd, log in, and click on the "Update My Information" link at the top of the page. Please update your user information to ensure an efficient production and billing process.

*Note to LaTeX users only - Our staff will ask you to upload a TEX file in addition to the PDF before the paper can be sent to typesetting, so please carefully review our Latex Guidelines [http://www.plosntds.org/static/latexGuidelines.action] in the meantime.

Best regards,

Eric Mossel, Ph.D.

Guest Editor

Samuel Scarpino

Deputy Editor

Reviewer's Responses to Questions

**Key Review Criteria Required for Acceptance?**

**Methods**

-Are the objectives of the study clearly articulated with a clear testable hypothesis stated?

-Is the study design appropriate to address the stated objectives?

-Is the population clearly described and appropriate for the hypothesis being tested?

-Is the sample size sufficient to ensure adequate power to address the hypothesis being tested?

-Were correct statistical analysis used to support conclusions?

-Are there concerns about ethical or regulatory requirements being met?

Reviewer #1: (No Response)

Reviewer #3: The edits made have provided clarity.

Much clearer / more integrated.

**Results**

-Does the analysis presented match the analysis plan?

-Are the results clearly and completely presented?

-Are the figures (Tables, Images) of sufficient quality for clarity?

Reviewer #1: (No Response)

Reviewer #3: changes made ensure clarity and a more transparent analysis (focusing on information related to the CCC studied and not all CCC)

**Conclusions**

-Are the conclusions supported by the data presented?

-Are the limitations of analysis clearly described?

-Do the authors discuss how these data can be helpful to advance our understanding of the topic under study?

-Is public health relevance addressed?

Reviewer #1: (No Response)

Reviewer #3: changes made have greatly improved paper.

**Editorial and Data Presentation Modifications?**

Reviewer #1: (No Response)

Reviewer #3: the changes made in this paper have made it clearer, easier to read and more transparent in terms of varied CCC structures and approaches and the realities in SL

Accept.

**Summary and General Comments**

Reviewer #1: Thanks for the opportunity to re-review this paper. The authors did a thoughtful revision which takes into account reviewers’ input from the original submission. Suggest the paper be accepted.

There are 2 small comments at this stage:

• Abstract -Line 18-19 : suggest the authors reframe the statement in the abstract on the aim of the paper from ‘why communities see CCCs in a positive light’ to something more open ended such as ‘assess community perceptions’ – similar to how this was framed later in the introduction line 107

• Discussion line 520 – suggest the authors remove reference to ‘statistical significance’ which is inappropriate for a qualitative assessment of this sort

Reviewer #3: Changes taken onboard have improved the integrity and transparency of the methodology and the realities within the SL response (making distinction between areas studies/ CCC communities included in the study vs. all CCC areas)

Overall, the paper has greatly improved and is much easier and clearer to read and follow

The clarifications on SDBs and the methodology has also improved.

PLOS authors have the option to publish the peer review history of their article (what does this mean?). If published, this will include your full peer review and any attached files.

Reviewer #1: No

Reviewer #3: No

---

## [Editor Report · Acceptance letter]

16 Jan 2020

Dear Dr Maat,

We are delighted to inform you that your manuscript, "Rural populations exposed to Ebola Virus Disease respond positively to localised case handling: evidence from Sierra Leone," has been formally accepted for publication in PLOS Neglected Tropical Diseases.

Best regards,

Serap Aksoy

Editor-in-Chief

Shaden Kamhawi

Editor-in-Chief
